# The genomic landscape of meiotic crossovers and gene conversions in *Arabidopsis thaliana*

Erik Wijnker[1,2†], Geo Velikkakam James[3†], Jia Ding[4], Frank Becker[1], Jonas R Klasen[3], Vimal Rawat[3], Beth A Rowan[5], Daniël F de Jong[1,6], C Bastiaan de Snoo[6], Luis Zapata[7], Bruno Huettel[8], Hans de Jong[1], Stephan Ossowski[7], Detlef Weigel[5], Maarten Koornneef[1,4], Joost JB Keurentjes[1]*, Korbinian Schneeberger[3]*

[1]Laboratory of Genetics, Wageningen University, Wageningen, Netherlands; [2]Department of Molecular Mechanisms of Phenotypic Plasticity, IBMP–CNRS, Université de Strasbourg, Strasbourg, France; [3]Department of Plant Developmental Biology, Max Planck Institute for Plant Breeding Research, Cologne, Germany; [4]Department of Plant Breeding and Genetics, Max Planck Institute for Plant Breeding Research, Cologne, Germany; [5]Department of Molecular Biology, Max Planck Institute for Developmental Biology, Tübingen, Germany; [6]Rijk Zwaan R&D Fijnaart, Rijk Zwaan, Fijnaart, Netherlands; [7]Bioinformatics and Genomics Programme, Center for Genomic Regulation (CRG) and Universitat Pompeu Fabra (UPF), Barcelona, Spain; [8]Max Planck Genome Center Cologne, Max Planck Institute for Plant Breeding Research, Cologne, Germany

*For correspondence: Joost. Keurentjes@wur.nl (JJBK); schneeberger@mpipz.mpg.de (KS)

†These authors contributed equally to this work

**Abstract** Knowledge of the exact distribution of meiotic crossovers (COs) and gene conversions (GCs) is essential for understanding many aspects of population genetics and evolution, from haplotype structure and long-distance genetic linkage to the generation of new allelic variants of genes. To this end, we resequenced the four products of 13 meiotic tetrads along with 10 doubled haploids derived from *Arabidopsis thaliana* hybrids. GC detection through short reads has previously been confounded by genomic rearrangements. Rigid filtering for misaligned reads allowed GC identification at high accuracy and revealed an ~80-kb transposition, which undergoes copy-number changes mediated by meiotic recombination. Non-crossover associated GCs were extremely rare most likely due to their short average length of ~25–50 bp, which is significantly shorter than the length of CO-associated GCs. Overall, recombination preferentially targeted non-methylated nucleosome-free regions at gene promoters, which showed significant enrichment of two sequence motifs.

## Introduction

Sexually reproducing organisms are thought to have an advantage over asexual species, as novel allele combinations can emerge after meiosis in each generation (*Otto and Lenormand, 2002*; *de Visser and Elena, 2007*). Besides randomly passing on one of the homologous chromosomes to the next generation, meiosis introduces intra-chromosomal recombination. In addition to potentially generating superior allele combinations, this shuffling of alleles avoids Muller's ratchet, in which asexual individuals accumulate deleterious mutations over time (*Muller, 1932*). The impact of meiotic recombination on the allele distribution among offspring, and consequently on the haplotype structure of whole populations, depends on the rate and positioning of recombination events, which are initiated by the induction of meiotic double strand breaks (DSBs).

**eLife digest** Most living organisms package their DNA into bundles called chromosomes. These chromosomes generally form pairs, with each chromosome in the pair containing the same number of genes. The genes also come in the same order, but the exact sequence of DNA bases within the genes can be different.

When sex cells—such as egg, sperm or pollen cells—are made, each pair of chromosomes is separated so that the each cell contains only half the normal number of chromosomes. However, before they are separated, the pairs swap lengths of DNA via recombination events. These can involve exchanging large chunks of the chromosomes: this is called a 'crossover'. Alternatively, short stretches of one chromosome can be replaced by the corresponding region from the other in the pair. When these 'non-crossovers' cause a change in the DNA sequence they are known as gene conversions.

Long-standing questions in the field of plant biology include: how common are gene conversions? How much DNA is typically exchanged? And where in the chromosomes do these events happen most? Now, Wijnker et al. have addressed these questions by focusing on the accurate detection of recombination events, with a special emphasis on gene conversions, in the plant biologist's favourite species: Arabidopsis.

Searching for recombination events is a challenge because, when piecing together an entire genome from lots of shorter stretches of DNA—typically called 'reads', it is easy to misplace some of the pieces. However, meticulous examination of these short DNA reads allowed Wijnker et al. to reliably identify gene conversions on a genome-wide scale. In Arabidopsis, gene conversion appears to be unexpectedly rare—with approximately one gene conversion detected per 140–240 non-crossovers. Recombination tends to occur in regions of the chromosomes where the DNA is only loosely packaged, is not heavily modified by the process of 'DNA methylation', and also near the start of genes. Furthermore, two specific sequences of DNA bases were identified that marked 'hot spots' in the chromosomes, where recombination happens more frequently.

Wijnker et al. suggest that the low number of gene conversions detected indicates that non-crossovers tend to exchange very short stretches of DNA. However, future research may point to additional mechanisms that explain the low incidence of gene conversion in Arabidopsis.

Meiotic DSBs are repaired through homologous recombination (HR), in which homologous sequences are used as repair templates (*San Filippo et al., 2008*). The broken strand invades a homologous chromosome and is repaired through HR repair intermediates like the D-loop and double Holliday junction. These intermediates are resolved either as crossovers (COs), leading to the reciprocal exchange of complete chromosome arms, or non-crossovers (NCOs). In both cases, heteroduplexes may arise from sequence divergence at the site of the strand invasion. When these are resolved, they can give rise to gene conversions (GCs), the non-reciprocal exchange of alleles between homologous non-sister chromatids. Such events typically lead to a 3:1 segregation of alleles among the four gametes of a single meiosis. GCs can be associated with COs (CO–GCs) or with NCOs (NCO–GCs) where they can introduce new alleles into an otherwise unchanged genetic background, depending on how the HR intermediate was resolved.

The number of GCs depends on the number of DSBs, polymorphism density, and on the tract lengths of HR repair intermediates (*Kauppi et al., 2009*; *Dooner and Martínez-Férez, 1997*). Though some general characteristics of meiotic recombination may be shared among sexually reproducing organisms, such as the 150–250 DSBs that are formed at the onset of meiosis in yeast (*Saccaromyces cerevisiae*) (*Weiner and Kleckner, 1994*; *Buhler et al., 2007*), mouse (*Moens et al., 1997*) and *A. thaliana* (*Vignard et al., 2007*; *Chelysheva et al., 2007*; *Sanchez-Moran et al., 2007*), the outcomes of meiosis with around 90, 27 and 10 COs per meiosis are profoundly different (*Mancera et al., 2008*; *Moens et al., 2002*; *Giraut et al., 2011*; *Salome et al., 2012*).

There are three recent reports assessing GC rates in *A. thaliana*. Lu et al. (2012) sequenced two meiotic tetrads at ~13x coverage (*Lu et al., 2012*) by making use of the quartet mutant (*qrt*), which allows for unordered tetrad analysis. They identified three CO–GCs and two NCO–GCs per meiosis. *Yang et al. (2012)* investigated NCO–GCs through whole-genome sequencing of 40 F$_2$ offspring,

reporting a 130-fold higher GC rate than *Lu et al. (2012)*. Finally, *Sun et al. (2012)* made use of fluorescent reporters (*Francis et al., 2007*) in a *qrt* mutant background to assess GC rates at transgenic loci, which when extrapolated to the whole-genome suggest a similar GC rate as *Lu et al. (2012)*.

We have sequenced the four products of each of 13 meiotic tetrads and 10 homozygous doubled haploid (DH) offspring obtained from heterozygous $F_1$ hybrids. We show that short read alignment-based artifacts, which result from structural differences between parental genomes, can greatly inflate the apparent incidence of GCs. In total, we identified over 200 recombination events to the nucleotide level. These data suggest that recombination does not seem to be affected by sequence divergence, but revealed their preference for open chromatin and a significant association to two sequence motifs. Aided by simulations we provide estimates on gene conversion rates and associated tract lengths. CO-associated GCs are at ~300–400 bp between 6–16 times longer than GCs resulting from NCOs (25–50 bp). The recovery of a ~80 kb transposition that through crossover recombination leads to duplications and deletions in the offspring underlines the potential of meiosis to not only create new allele combinations but also to cause copy-number alterations of transposed sequences.

## Results

### Sequencing and genotyping of meiotic tetrads

We constructed $F_1$ hybrids of the *A. thaliana* accessions Columbia (Col) and Landsberg *erecta* (L*er*) in the *quartet1 (qrt1)* −/− background (*Preuss et al., 1994*) and crossed individual pollen tetrads to a male sterile EMS mutant of the Cape Verde Island (Cvi) accession as a female receptor. This generated four heterozygous offspring, each composed of one haploid genome of Cvi and one haploid recombinant Col-L*er* genome, referred hereafter to as a complete tetrad (*Figure 1A*, 'Materials and methods'). The 20 genomes of five complete tetrads were sequenced at an average coverage of 54x, whereas eight other complete tetrads were sequenced at lower average coverage of 14x, amounting to a total of 52 sequenced tetrad offspring (*Supplementary file 1A*). In addition we sequenced the genomes of all three parental accessions and performed standard resequencing analyses, which yielded a list of 269,842 high quality bi-allelic single nucleotide polymorphism (SNP) markers that distinguish the Col and L*er* alleles (average coverage 56x, *Supplementary file 2A*, 'Materials and methods') (*Ossowski et al., 2008*).

To reconstruct the recombinant Col-L*er* chromosomes of individual tetrad offspring, we first assigned either a Col or L*er* genotype to every marker in the genome and subsequently merged blocks of consecutive markers with the same genotypes for the identification of crossover break points. This block-merging step occasionally merged interspersed markers with different genotypes. While this efficiently removes all genotyping errors, it will not allow for the identification of any GCs. The visualizations of these stretches were used to create graphical genotypes (e.g., in *Figure 1B*).

Since COs lead to reciprocal exchange, complete tetrads allow the reconstruction of all CO recombination events of a single meiosis. At least one CO was identified on each chromosome, with an average of 10.1 COs per meiosis (*Figure 1B*, *Figure 1—figure supplement 1*). The larger chromosomes one, three and five showed more COs than chromosomes two and four, which is in agreement with earlier reports on CO frequency in male meiosis in *A. thaliana* (*Giraut et al., 2011*; *Wijnker et al., 2012*) (*Supplementary file 1B*). In total, we recovered 131 COs throughout all 13 tetrads (*Supplementary file 1C*).

### NCO identification in tetrad offspring

Unlike COs, NCOs can only be identified when they result in GCs. NCO–GCs can be detected by searching for L*er* alleles in regions inherited from Col, and *vice versa*. We distinguished between two different types of NCO–GCs. The first type can be detected at markers where the expected allele is isogenic to the recipient parent (Cvi). Here, NCO–GCs alter a homozygous into a heterozygous genotype (type-1 NCO–GCs). The second type is detected when the expected allele is different from the Cvi allele, and NCO–GCs change a heterozygous genotype into a homozygous genotype (type-2 NCO–GCs, *Figure 2—figure supplement 1*). As the footprint for type-2 NCO–GCs is the absence of one of the parental alleles, high sequencing depth is required to be confident in assigning a null-allele. In a previous study, this was addressed by removing all type-2 NCO–GCs from the analysis, effectively removing around 50% of the markers (*Lu et al., 2012*).

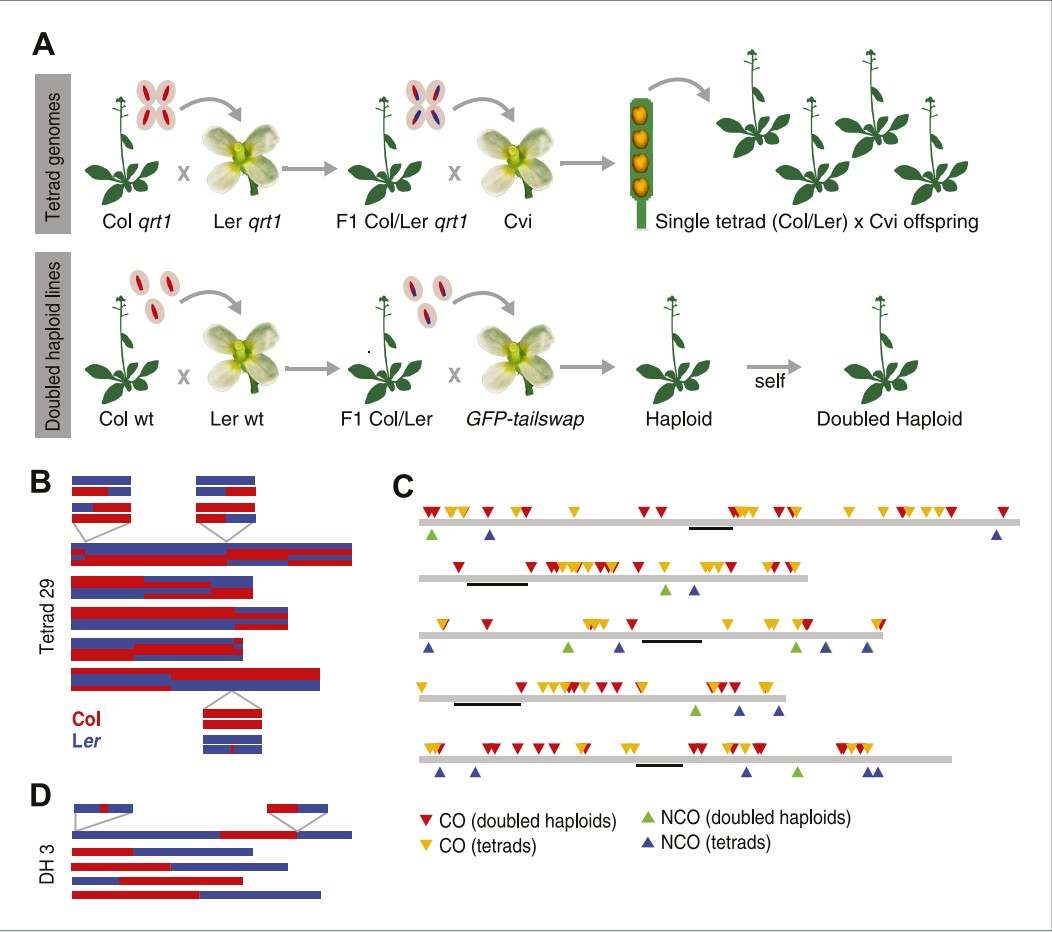

**Figure 1**. Experimental design and summary of recombination events within 62 recombinants. (**A**) 13 complete tetrads were generated by crossing *qrt1* in a Col background to *qrt1* in a L*er* background, and then using single pollen tetrads from the F$_1$ hybrids to fertilize a Cvi male sterile pollen receptor. Tetrad offspring are heterozygous, with one recombinant Col-L*er* genome and one Cvi genome. 10 DH lines were generated by crossing wildtype Col-L*er* F$_1$ hybrids to the *GFP-tailswap* haploid inducer. One round of selfing generated doubled haploids. (**B**) Example of graphical genotypes of all five Col-L*er* recombinant chromosomes of all four offspring of a complete tetrad. Col regions are shown in red, L*er* regions in blue. The homologous chromosome, which is inherited from Cvi, is not shown. The three enlarged regions show a CO–GC, a CO without GC and a NCO–GC (clockwise, starting at the upper left corner). (**C**) All recombination events identified in this study. Different recombination types are labeled with different colors. Centromere positions are indicted with black lines. (**D**) Example of the graphical genotypes of the five chromosomes of one DH line (Col regions are shown in red, L*er* regions in blue). The two enlarged regions show a NCO–GC (left) and a CO without GC (right).

The following figure supplements are available for figure 1:

**Figure supplement 1**. Graphical genotypes of all 13 complete tetrads.

**Figure supplement 2**. Graphical genotypes of 10 DH lines.

Precise identification of NCO–GCs relies on accurate detection of homozygous and heterozygous states at polymorphic markers, which in turn depends on two criteria: a reliable short read alignment-based allele frequency threshold for distinguishing between homozygous and heterozygous genotypes, and a sufficient number of aligned sequence reads. In order to define the first, we summarized short read alignment-derived frequencies of the parental allele at all type-1 and type-2 marker positions. Fitting beta distributions to the observed allele frequency distributions allowed for the identification of a global, coverage-dependent cutoff that distinguishes between heterozygous and homozygous genotypes (*Figure 2—figure supplement 2*, 'Materials and methods').

To establish a minimum coverage threshold, we reasoned that insufficient coverage will lead to increased detection of (false-positive) NCO–GCs, but once a critical coverage is met, higher coverage requirements will not further impact the frequency of NCO–GCs candidate predictions. We calculated the frequency of NCO–GCs (defined as the ratio of converted markers divided by all markers) for a wide range of minimal coverage thresholds (*Figure 2—figure supplement 3*). At coverage levels of at least 50x, the frequency of NCO–GCs remained stable, and this sequence depth was subsequently used as the minimal threshold.

Combining both the allele frequency and the coverage thresholds, we were able to confidently assign genotypes to 2,477,241 markers within the five deeply sequenced complete tetrads. Among these, 1,359 revealed putative NCO–GCs.

## Genome rearrangements can lead to false-positive NCO–GC calls

Type-1 and type-2 NCO–GC classification is a consequence of allele sharing with the recipient Cvi genotype. Because Col and L*er* alleles segregate in an equal (2:2) ratio among the four tetrad offspring, both types of NCO–GCs are expected to occur at similar numbers. However, we identified a striking overrepresentation of type-1 NCO–GCs (*Figure 2—figure supplement 4*). As mentioned, type-1 NCO–GCs introduce heterozygous genotype calls at markers where homozygous genotype calls would be expected. Such patterns of putative heterozygous alleles can also be introduced by erroneously aligned reads. By manually screening short read alignments, we found a large fraction of putative NCO–GCs residing in regions in which the parental lines had been assigned heterozygous genotypes. Since all three parental lines are highly inbred, and hence homozygous throughout their genomes, the presence of heterozygous marker calls can only be explained by misaligned reads that result from unknown rearrangements or repeats. This is especially critical for L*er* and Cvi, where genome information is more limited than for the reference accession Col. *Figure 2A* shows an example where three haplotypes appear to be present at a single locus when the short read data of L*er* are aligned against the Col reference genome. This specific marker was used to assign NCO–GCs in two earlier studies (*Yang et al., 2012*; *Lu et al., 2012*). To prevent such erroneous calls, we redefined the initial set of markers by excluding all markers near regions with evidence of putative duplications ('Materials and methods').

Repeating the analysis with the filtered set of markers still revealed apparently false positive NCO–GC calls. Most remarkable was an ~80 kb region on chromosome three, in three out of five independent tetrads, which harbored multiple closely linked putative type-2 NCO–GCs, but not a single type-1 NCO–GC. Thus, all these putative NCO–GCs were homozygous genotypes, which suggested another source of error. Intriguingly, for each of these tetrads, one other offspring was found with a similar pattern in the same region, but in contrast these offspring featured only putative type-1 NCO–GCs (heterozygous genotypes). Manual inspection of short read alignments and de novo assemblies of L*er* revealed a so-far undescribed large-scale rearrangement between Col and L*er* (*Figure 2B*, *Figure 2—figure supplement 5*). This rearrangement encompasses two closely linked transpositions, which together relocated ~40 kb (including six genes) from ~22.53–22.57 Mb of the reference genome to 17.36 Mb on chromosome 3 in the genome of L*er*. Additionally, the region adjacent to the transposed sequences (between 22.51 Mb and 22.59 Mb on chromosome 3) appeared to be mostly absent in L*er*. The segregation of this complex region in combination with a CO in between the two insertion sites of the transpositions resulted in either a duplication or a deletion of the transposed sequence within individual offspring genomes (*Figure 2C*). As a consequence, this unexpected copy number variation introduced false NCO–GC calls (*Figure 2D*). Among all 13 tetrads (including the eight shallow sequenced tetrads), there are seven that show a recombination event between the transposed regions, suggesting that an ~80 kb duplication/deletion is present in about half of all Col-L*er* gametes.

An earlier report highlighted the region at ~22.5 Mb as hotspot for double recombination (*Yang et al., 2012*). Intriguingly, each of the genomes with a putative double recombination event at this position within the graphical genotypes also featured a CO between the insertion sites, which leads to the above-mentioned copy number variations and misleading genotypes (*Figure 2E*).

To remove these and similar patterns we utilized the presence of the Cvi alleles. We assigned individual read pairs to one of the three parental genotypes whenever closely linked polymorphisms allowed distinguishing all three parental alleles. All marker loci where short reads aligned from all parental lines, were removed from further analysis. In addition, we removed markers

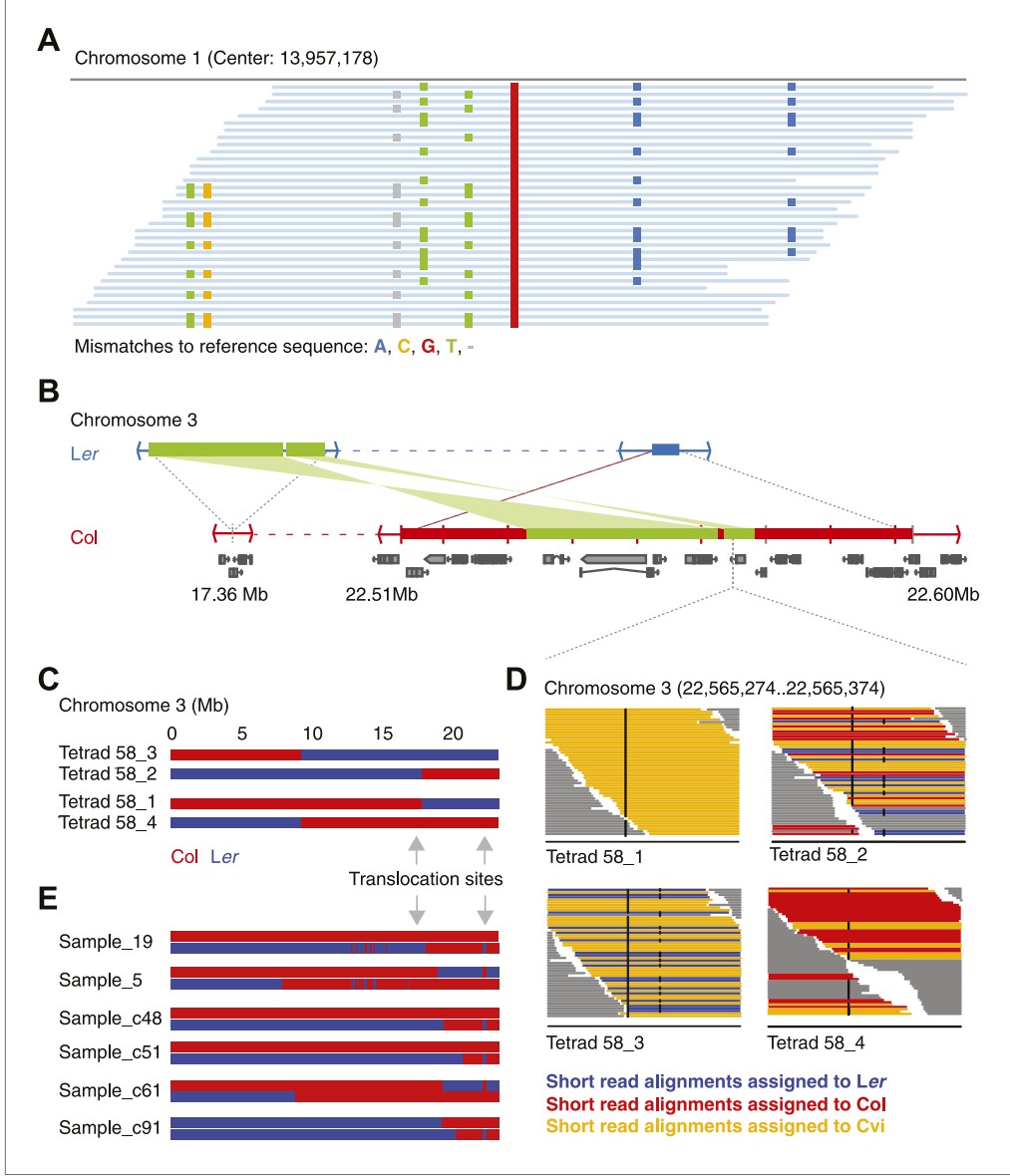

**Figure 2**. Identification of gene conversions is complex because of unknown duplications and transpositions in the *A. thaliana* genome. (**A**) Short read alignments of L*er* against the reference sequence at position 13,957,178 on chromosome 1. Individual reads are shown as blue lines, while mismatches between reference sequence and short reads are colored according to mismatch types. Three distinct L*er* haplotypes align to this region, indicating that this sequence is present in triplicate in the L*er* genome. As L*er* is homozygous, at least two haplotypes were not aligned to their respective origin. (**B**) The genomic landscape of the two insertion sites of an ~80 kb transposition between L*er* and Col. Blue and red boxes mark sequences that are unique to L*er* and Col respectively. Green boxes highlight the transposed (and inverted) DNA. Genes annotated in Col are shown in grey. (**C**) Graphical genotypes of chromosome 3 of the four genomes of tetrad 58 (Col-derived genomic regions are shown in red, L*er* regions in blue). Cvi sequences are not shown. Grey arrows indicate the insertion sites of the transposition illustrated in **B**. Tetrad 58_1 and tetrad 58_2 formed a crossover between these insertion sites. As a result, tetrad 58_1 lost all transposed sequences, whereas in tetrad 58_2 the transposed DNA is duplicated. (**D**) Short read alignments of all four genomes of tetrad 58 to chromosome 3 at positions 22,565,274 to 22,565,374, which overlap the transposed DNA. This region includes two closely linked SNPs that can distinguish all three parental alleles (black dots indicate mismatches to the reference sequence). The reads that can be assigned to one of the three parents are shown by different colors. Tetrad 58_1, which lost the transposed DNA, shows the absence of Col and L*er* derived reads, whereas tetrad 58_2, which inherited both transposed regions, shows the presence of both Col and L*er* alleles in this region. (**E**) Redrawing of the graphical genotypes of six Col-L*er* F₂ offspring, as presented in the appendix of *Figure 2. Continued on next page*

*Figure 2. Continued*

***Yang et al. (2012)***. These offspring experienced a putative double CO, co-localizing with the Col insertion site of above-mentioned transposition. Note that in all six $F_2$ offspring, at least one of the recombinant chromosomes features a CO between the transposition sites. This suggests that the annotated double recombinations are not real, but that the observed patterns originate from copy number variation due to transposed DNA.

The following figure supplements are available for figure 2:

**Figure supplement 1**. Graphical illustration of short read alignments at type-1 and type-2 markers revealing no NCO–GCs (two loci at the left) and the same loci revealing type-1 and type-2 NCO-GCs (right).

**Figure supplement 2**. Observed allele distribution throughout all deeply sequenced tetrad genomes.

**Figure supplement 3**. NCO–GC frequency per marker per meiosis measured in the five deeply sequenced tetrads, using a range of minimal coverage thresholds and three different marker sets.

**Figure supplement 4**. The number of putative type-1 and type-2 NCO–GCs in all 20 deeply sequenced tetrad offspring using different marker sets.

**Figure supplement 5**. Transposed sequences on *A. thaliana* chromosome 3.

**Figure supplement 6**. NCO–GC frequency per marker per meiosis measured in 10 recombinant DH lines at increasing minimal coverage thresholds.

where the local sequence divergence complicated the alignment of L*er* short reads to the reference sequence, since such regions can interfere with the allele frequency calculation ('Materials and methods').

With the final set of 137,339 markers, we genotyped all 20 deeply sequenced tetrad offspring for the presence of putative NCO–GCs at markers that were not found to be involved in CO-associated gene conversion (***Supplementary file 2B***, 'Materials and methods'). We were able to confidently assign genotypes to 1,092,055 loci within the 20 individuals, revealing 10 putative NCO–GCs that were supported by a total of 12 converted markers. PCR-based sequencing confirmed the 3:1 segregation of seven NGO–GCs (based on seven markers), but rejected three putative NCO–GCs (based on five markers) because there was a 2:2 segregation of the alleles which was not apparent in the short read sequencing data (***Supplementary file 1D***, 'Materials and methods'). The false-positive predictions may result from incomplete filtering against complex sequence differences, for which no closely linked markers were available to distinguish the reads of all three parents.

## Doubled haploid lines as independent controls for NCO–GC detection

A reliable calculation of NCO–GCs is complicated by the heterozygous nature of our samples and stringent filtering may have reduced the number of identified GCs. In order to confirm our analysis of NCO–GCs in an independent experiment, we resequenced the homozygous genomes of 10 doubled haploid offspring of Col-L*er* $F_1$ hybrids (average coverage 49x, ***Supplementary file 1A***) that we randomly selected from an previously generated DH population (***Wijnker et al., 2012***) ('Materials and methods'). These were generated by crossing the heterozygous $F_1$ (as male) to the *GFP-tailswap* haploid inducer (***Ravi and Chan, 2010***) to produce haploid plants with recombinant genomes, which were selfed to give rise to diploid, doubled haploid (DH) lines (***Wijnker et al., 2012***) (***Figure 1A***, ***Figure 1C***). The genomes of the DH lines have blocks of homozygous regions derived either from Col or L*er*, which resulted from recombination events during meiosis in the $F_1$ parent.

In addition, we resequenced the parental lines of the DH lines and defined a set of 438,915 high quality markers (average coverage 58.9, ***Supplementary file 2C***, 'Materials and methods'). Genotyping and identification of consecutive blocks of the same genotype were performed in analogy to the tetrad sample analysis ('Materials and methods'). This revealed 60 COs in total (***Supplementary file 1E***). Only one of these CO sites revealed the presence of a GC, which is expected, since single gametes cannot reveal the complete picture of CO–GCs (***Qi et al., 2009***) (***Figure 1D***, ***Figure 1—figure supplement 2***).

In order to test if the haploid life cycle during the generation of DH lines interferes with the assignment of NCO–GCs, we searched for spontaneous mutations in all ten DH lines. To this end, we analyzed 743,440,307 positions across all ten genomes, for which we had sufficient, non-ambiguous short read alignments and identified eight spontaneous mutations ('Materials and methods'). From this, we estimated a spontaneous mutation rate of $1.1 \times 10^{-8} \pm 1.0 \times 10^{-8}$ per site per generation, which is slightly higher than the estimated mutation rate of $7.0 \times 10^{-9}$ for sexually reproducing plants (*Ossowski et al., 2010*). Like for the spontaneous mutation rates, we observed an enrichment of transitions (n = 6) over transversions (n = 2). An increased mutation rate in haploids could result from the absence of a homologous repair template in haploid (G1) cells that might lead to increased repair through the presumably more error-prone pathway of non-homologous end-joining (*Gorbunova and Levy, 1997*; *Mao et al., 2008*). Alternatively, the process of uni-parental genome elimination (*Sanei et al., 2011*) could potentially cause mutagenic stress. Even if spontaneous mutation rates are increased in the DH lines, there are on average no more than two mutations per DH genome, which is by far not large enough to interfere with the accurate identification of GCs, which act on existing variation only.

To assess a minimal coverage threshold to identify putative NCO–GCs for the DH lines, we used the same allele frequency cutoffs as for the analysis of the tetrad samples and calculated the conversion frequency for a series of minimal coverage thresholds. From a minimal coverage of 10 onwards, the frequency of NCO–GCs levels off (*Figure 2—figure supplement 6*). Using this value as threshold, we genotyped 438,915 markers in each of the 10 DH lines ('Materials and methods') and could confidently assign genotypes to 3,672,610 loci. Among these we identified 10 putative NCO–GC tracts (converting 13 markers in total). PCR-based sequencing of these loci confirmed nine NCOs (that converted a total of 10 markers), but rejected one putative NCO–GC based on three converted markers (*Supplementary file 2*).

## The endogenous rate of gene conversion in *A. thaliana*

We estimated the frequency of NCO–GCs in the five deeply sequenced tetrads and 10 independent DH lines as $5.9 \times 10^{-6} \pm 6.1 \times 10^{-6}$ and $2.7 \times 10^{-6} \pm 2.7 \times 10^{-6}$ per site per meiosis respectively, numbers that are in very close agreement. Combining both experiments we estimated a frequency of $3.6 \times 10^{-6} \pm 2.7 \times 10^{-6}$ for NCO–GCs per site per meiosis, a frequency that is three orders of magnitude higher than the spontaneous mutation rate (*Ossowski et al., 2010*). In addition, we calculated the frequency of CO–GCs based on the tetrad data and estimated this rate at $7.8 \times 10^{-6} \pm 5.4 \times 10^{-6}$ per site per meiosis, which is not significantly different from the NCO–GC frequency. See *Figure 1D* for a spatial overview.

## Sequence divergence within CO-associated GCs

To obtain a more detailed view on recombination sites, we reconstructed the sequences around GC sites from the tetrad samples by manually inspecting and locally assembling short reads to obtain the full sequence within conversion tracts. We focused on a subset of 71 COs for which sufficient sequencing information was available in both gametes. All 71 CO sites have been visualized in *Supplementary file 3*. Of these COs 62.0% (n = 44) showed the presence of associated GCs (*Supplementary file 1F,G*). All CO-associated conversion tracts (COCTs) except one showed co-conversion of adjacent polymorphisms (*Schultes and Szostak, 1990*), indicating that all alleles in the COCT segregated in a 3:1 ratio. This observation compares well to yeast, where co-conversion of alleles prevails (*Schultes and Szostak, 1990*; *Stahl and Foss, 2010*). We observed 23 and 20 co-conversions to Ler and Col, respectively, which is not significantly different (p value 0.64 [$\chi^2$-test]).

Occasionally, sequence divergence within the COCTs was large. We observed COCTs with multiple deletions and insertions of up to 18 bp in length. In one of them, Ler is different from Col at almost one fifth of all positions throughout the COCT. Since homologous recombination requires homology for strand invasion, we speculated that overall COCTs should prefer regions with higher similarity. The distribution of sequence identity at COCTs was not found to be significantly different from randomly sampled regions from the chromosome arms with similar lengths. However, around 3.9% of these regions featured less than 5% confidently aligned positions, when aligning short reads of Ler against the reference sequence. Such regions, which nearly completely miss any homology, were not targeted by any recombination according our data (*Figure 3A*). Nevertheless, reliable identification of a true lower border for the tolerance for sequence divergence at CO sites will be difficult to establish. This would require many more CO events, but also more precise information on the positioning of DSBs and strand invasion, since it is possible that recombination events are initiated adjacent to their resulting conversion tracts.

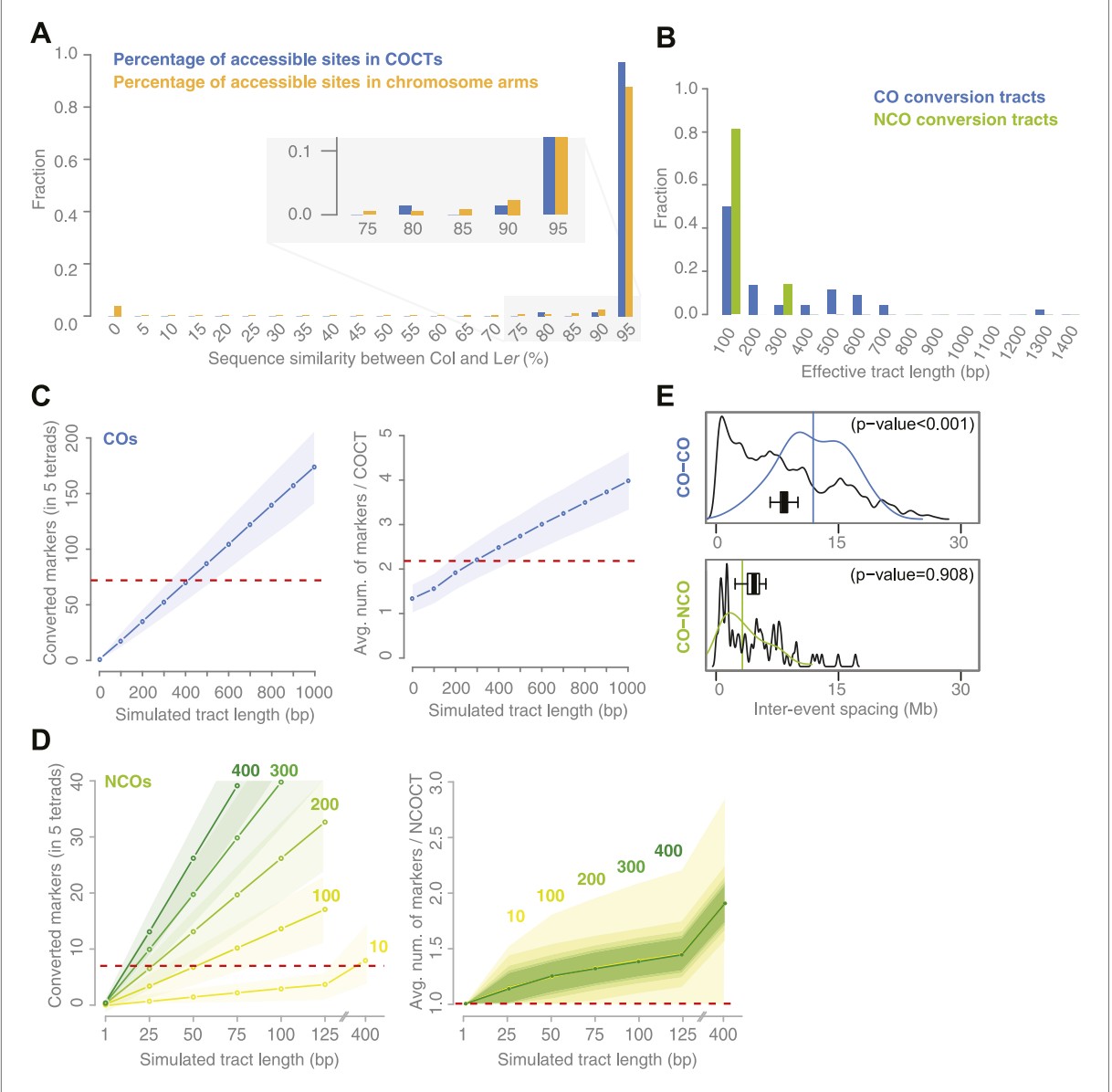

**Figure 3**. Sequence similarity at recombination sites, length differences between COCTs and NCOCTs and crossover interference. (**A**) The percentage of confidently aligned positions within the resequencing of L*er* was used as proxy for local sequence similarity. The percentages at COs were compared to a background distribution based on random sampling in non-peri-centromeric regions. (**B**) Comparison of COCT and NCOCT lengths. COCTs are significantly longer. (**C**) Repeatedly simulating (n = 10,000) sets of 52 COs randomly placed in the non-peri-centromeric regions predicted the average number of converted markers throughout all COCTs (left) as well as the average number of co-converted markers within a single COCT (right) given a fixed, simulated tract length (blue lines show the average values, shaded regions indicate standard deviations). The dashed red lines indicate the observed values for the real number of converted markers and average number of markers that co-converted. The intersections of observed and simulated numbers suggest an average COCT length of ~300–400 bp. (**D**) Estimation of NCOCT length, as shown for COCTs in **C**. As the absolute number of NCOs is not obvious, we simulated a range of different NCO numbers (green to yellow colors indicate different numbers of DSBs per meiosis, of which half are simulated not to restore the original allele). Assuming a NCOCTs length of ~400 bp (as we estimated for COCT length), only five NCOs would be formed per meiosis (left), however in this scenario the average number of co-converted markers would deviate drastically from the observed value (right). (**E**) The density distribution of distances between neighboring CO and between neighboring CO and NCO events reveals differences in inter-event spacing. The observed distances between recombination events (colored), with the average inter-event distance shown as a vertical line, are compared to randomized inter-event distances (black lines) measured after randomizing the labels of the existing tetrads. The boxplots show the distribution of the means of each randomization. The CO–CO distances are significantly longer than random distances due to crossover interference. Interference between COs and NCOs could not be detected.

To assess sequence divergence at NCO–GCs, we used the NCO–GCs recovered in the deeply sequenced tetrads and added another 11 NCO–GCs identified in the same tetrads. These additional GCs did not pass our final filters, but have all been confirmed by PCR. In total, we identified 18 converted polymorphisms, which could be assigned to 14 distinct NCO conversion tracts (NCOCTs) (*Supplementary file 1H*). Ten of these NCOCTs consisted of a single polymorphism and the remaining four of two converted polymorphisms. Such short NCOCTs cannot be used to distinguish between NCOCTs resulting from co-conversion and more complex tracts of alternating 3:1 and 2:2 segregating markers, as the detection of complex conversions requires NCOCTs with at least three consecutive markers.

## COs have longer gene conversion tracts than NCOs

The length distributions of observed CO and NCO conversion tracts are significantly different (p value 0.03 [*t* test], *Figure 3B*). Estimating the actual GC tract lengths from the observed GC tract lengths is difficult if the observed tracts are short and the marker density is relatively low. In particular for NCOCTs, where a majority of observed tracts had a length of 1 bp, tract length estimations are tenuous. We therefore performed simulations to determine the expected total number of converted markers within the five deeply sequenced tetrads and the average number of markers within a single conversion given different lengths of simulated conversion tracts.

For COCTs, we simulated 10,000 sets of 71 randomly placed COCTs of increasing conversion tract lengths from 100 bp to 1 kb and calculated the resulting total number of converted markers among all 71 COCTs and the average number of converted markers per tract. We found that a simulated length of ~400 bp corresponds to an average number of 69.9 converted markers, which is very close to the 72 converted markers observed in the tetrad samples. An average of 2.2 converted markers per tract corresponds to a simulated COCT length of ~300 bp (*Figure 3C*). Differences between these two estimates might result from the fact that we simulated a fixed COCT length, while actual COCTs might feature variability in their lengths. From this we estimated that COCTs are on average ~300–400 bp in length.

We repeated this simulation for NCOCTs. In contrast to simulations for COCTs, simulating NCOCT placement is hampered by the lack of knowledge of how many NCOs were present. It is likely that not all NCOs lead to GC, either due to restoration of the original allele as was assumed for half of the NCOs in yeast (*Mancera et al., 2008*), or because they occur in regions without polymorphisms. We therefore simulated different NCO rates within the range of reported DSBs per meiosis (*Vignard et al., 2007*; *Chelysheva et al., 2007*; *Sanchez-Moran et al., 2007*). We further assumed that half of them are restored to the original alleles and thus only 50% of them have the potential to lead to conversions (green to yellow colored lines in *Figure 3D*). These simulations suggested that 100 DSBs per meiosis with a tract length of 50 bp would lead to the detection of 6.7 converted markers on average, which is close to the observed number of 7. Likewise, simulating 200 DSBs per meiosis with a tract length of 25 bp would lead to a similar number of converted markers (average number of conversions was 6.5) (left side of *Figure 3D*).

An alternative hypothesis is that a substantial fraction of meiotic DSBs is repaired through the (identical) sister chromatid and thus does not lead to GC. In order to test this, we simulated a range of very small numbers of meiotic DSBs with a tract length of 400 bp (our estimate of COCT length) and found that no more than 10 DSBs per meiosis repaired through the homologous chromatid would be sufficient to convert as many markers as we had observed in our data (*Figure 3D*). However, simulated tracts of ~400 bp co-converted significantly more markers per CT as we observed in the real conversions (right side in *Figure 3D*).

Moreover, short NCOCTs lengths (averaging at 25–50 bp) would be in agreement with the low recovery rate of NCOs in studies on plant meiosis, and would concur with reported estimates of meiotic DSBs (*Vignard et al., 2007*; *Chelysheva et al., 2007*; *Sanchez-Moran et al., 2007*). It is important to note that short NCOCTs do not mutually exclude the possibility that some of the DSB are repaired through the sister chromatid.

## Interference acts between COs but not between COs and NCOs

Genetic interference is the phenomenon that the distance between adjacent recombination events along the chromosome axis is longer than expected under random placement. Interference between COs has been extensively studied in many different species, but reports on interference between COs and NCOs are scarce (*Berchowitz and Copenhaver, 2010*; *Mancera et al., 2008*). To study

interference, we calculated distances between adjacent COs and between NCOs and neighboring COs in our tetrad data set (*Figure 3E*). These were compared to expected distances between recombination events after randomizing the composition of tetrads by random sampling from all tetrad offspring. This preserved the non-uniform distribution of recombination events along the chromosome axis, while removing dependency between them as previously proposed by *Mancera et al. (2008)* ('Materials and methods'). Observed inter-CO distances (average distance of 11.9 Mb) were significantly longer than expected based on randomized tetrads (average distance of 8.3 Mb; p value <0.01 [permutation test]), confirming CO–CO interference. Observed CO–NCO distances fell within the expected range (average distance of NCOs: 3.2 Mb, average distance in randomization: 4.5 Mb, p value 0.91 [permutation test]), evidencing no significant difference from random placement. This either suggests that interference acts between COs only, or that the level of interference is so low we could not detect it.

## Recombination targets gene promoters with low levels of DNA methylation

The placement of recombination events is a complex interplay between various genomic features (e.g., see *Pan et al., 2011*). The precise localization of 67 COCTs and NCOCTs allowed us to investigate what local dependency may influence recombination localization in *A. thaliana*. Since conversion tracts might form only to one side of a DSB, we included 500 bp of flanking sequence on either side of the conversion tracts.

Recombination sites in animal and fungal genomes have been shown to correlate with high GC content, a presumed effect of biased gene conversion, in which AT/GC mismatches, are preferentially repaired to CG basepairs (*Pan et al., 2011*; *Duret and Galtier, 2009*; *Pessia et al., 2012*). We tested whether recombination events in *Arabidopsis thaliana* may be correlated with an elevated GC content. The GC content in the 67 conversion tracts was significantly lower than the genomic background (*Figure 4A*). This suggests that recombination in *A. thaliana* may be biased towards AT rich regions.

In *A. thaliana*, CG, CHG and CHH methylation show enrichment in peri-centromeric regions, while the high levels of gene body methylation comprise almost exclusively CG methylation (*Cokus et al., 2008*; *Lister et al., 2008*). A string of recent papers investigated the role of DNA methylation in *A. thaliana* recombination (*Yelina et al., 2012*; *Melamed-Bessudo and Levy, 2012*; *Colomé-Tatché et al., 2012*; *Mirouze et al., 2012*). They concordantly report a general increase in recombination in the chromosome arms when DNA methylation levels are suppressed. Consistently, we found that the percentage of methylated DNA for all types of DNA methylation, CG, CHG and CHH methylation, was significantly lower at recombination sites as measured within somatic tissue (mature rosette leaves) suggesting that local levels of DNA methylation interfere with recombination (*Figure 4B*) (*Stroud et al., 2012*).

Previous reports suggested that *A. thaliana* recombination hotspots co-localize with transposable elements (*Horton et al., 2012*) and that recombination events are enriched in peri-centromeric regions (*Yang et al., 2012*). This seems to be at odds with the avoidance of methylated DNA as our data suggested. We estimated the placement of recombination with respect to gene annotation (*Figure 4C*). Gene promoters and gene ends are significantly overrepresented among our recombination sites (p values 0.03 and 0.02 respectively [permutation tests]). Gene body regions are underrepresented among our recombination sites, but their underrepresentation is not significant. As our DNA methylation analysis suggested, we do not find evidence for an increase in recombination at transposable elements.

While we observed less recombination in genes than expected for random placement, the largest fraction of recombination events did however occur in genes. This implies that recombination frequently leads to the formation of new allelic variants of genes. Of 71 CO events in the tetrad samples, two generated new alleles of genes encoding putative 'hybrid proteins' in both of the respective gametes. Likewise five of the 60 CO events in the DH lines overlapped with genes, resulting in the generation of putative new allelic variants of genes. This is most likely a severe underestimate, as we only used available markers and might have missed some of the remaining polymorphisms, which could give rise to new variants of genes. Even though NCOs are relatively rare events, we found four NCOs that resulted in non-synonymous substitutions.

## DNA-motifs associated with recombination sites

It has been shown for a variety of species that putative DNA binding motifs may be associated with recombination hotspot activity (*Baudat et al., 2010*; *Horton et al., 2012*; *Choi et al., 2013*; *Comeron et al., 2012*; *Myers et al., 2008*). In *A. thaliana*, it was only possible so far to analyze ancestral

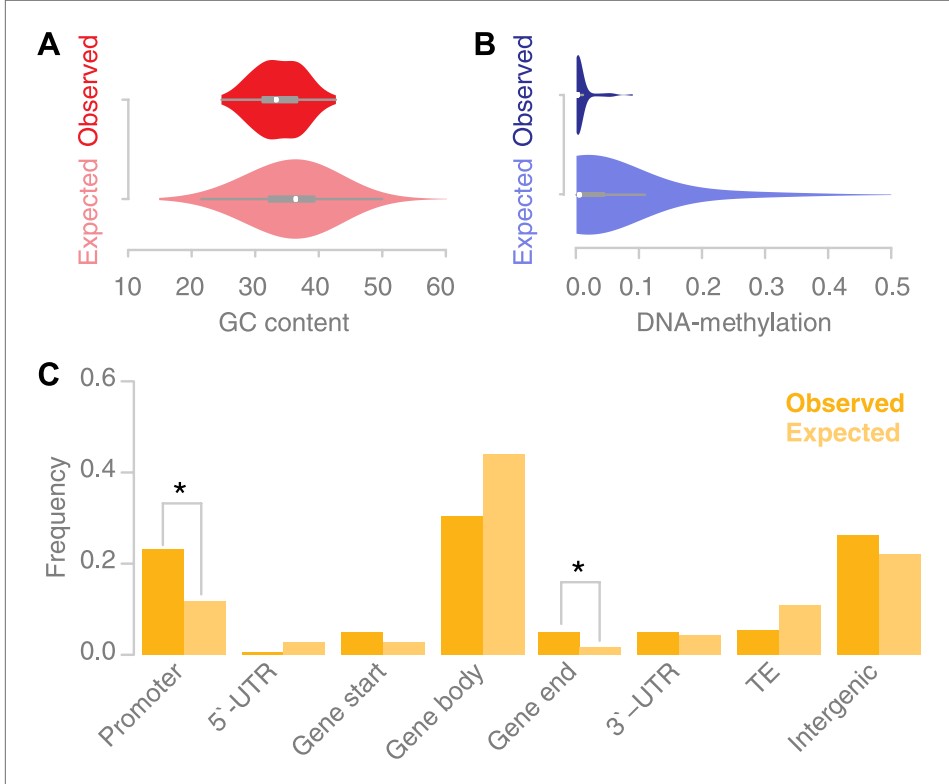

**Figure 4**. Recombination sites are enriched for un-methylated, AT rich promoter regions. (**A**) The GC content was calculated for 67 conversion tracts (including 500 bp of flanking sequence, top) and compared to a background distribution of 5,000 equally sized random locations sampled from non-peri-centromeric regions (bottom). Mean and variance are significantly different (mean: p value $2.4 \times 10^{-5}$ [$t$ test]; variance: p value 0.02 [Levene's test]). (**B**) The level of DNA methylation at recombination sites (top) was estimated based on bisulfite-treated DNA sequencing of mature rosette leaves (**Stroud et al., 2012**). DNA methylation at recombination sites (top) is significantly lower as compared to 5,000 equally sized random regions selected from chromosome arms (bottom) (p value $2.2 \times 10^{-16}$ [$t$ test]). (**C**) Associating recombination sites and gene annotations reveals a significant enrichment of recombination sites in promoters and gene ends. Promoters were defined as 500-bp regions upstream of transcription start sites, gene ends as the last 200 bp of a gene. The background distributions were estimated by randomly sampling from non-peri-centromeric regions.

recombination hotspots identified via linkage-based methods. Two recent studies reported on a collection of A- and CT-rich motifs in Arabidopsis, which are enriched at ancestral recombination hotspots (**Horton et al., 2012**; **Choi et al., 2013**). We searched for enriched sequence motifs at recombination sites and found two distinct motifs, which were significantly overrepresented in the set of 67 conversion tracts using the motif identification program MEME ('Materials and methods'). The first was a palindromic GAA/CTT microsatellite, present in ~51% (34 out of 67) of sequences (**Figure 5A**). The second was a poly-A homopolymer, present at ~76% (51 out of 67) of the recombination sites. To verify our findings we assessed the occurrences of these motifs within random genomic regions selected from the chromosome arms and found that both motifs occur at significantly higher frequencies at recombination sites as compared to genomic background random regions in the chromosome arms (GAA/CTT: p value 0.01; poly-A: p value $5.0 \times 10^{-7}$, [permutation tests], **Figure 5A**, 'Materials and methods').

## Recombination in *A. thaliana* targets constitutively open chromatin

Poly-A motifs are common in eukaryotic genomes and are known to prevent nucleosomes from binding DNA. Nucleosome-free regions are known to be targeted by the recombination machinery in yeast (**Pan et al., 2011**; **Wu and Lichten, 1994**). In mouse, conversely, meiotic DSBs apparently target nucleosome bound regions guided by the methyltransferase PRMD9 (**Smagulova et al., 2011**).

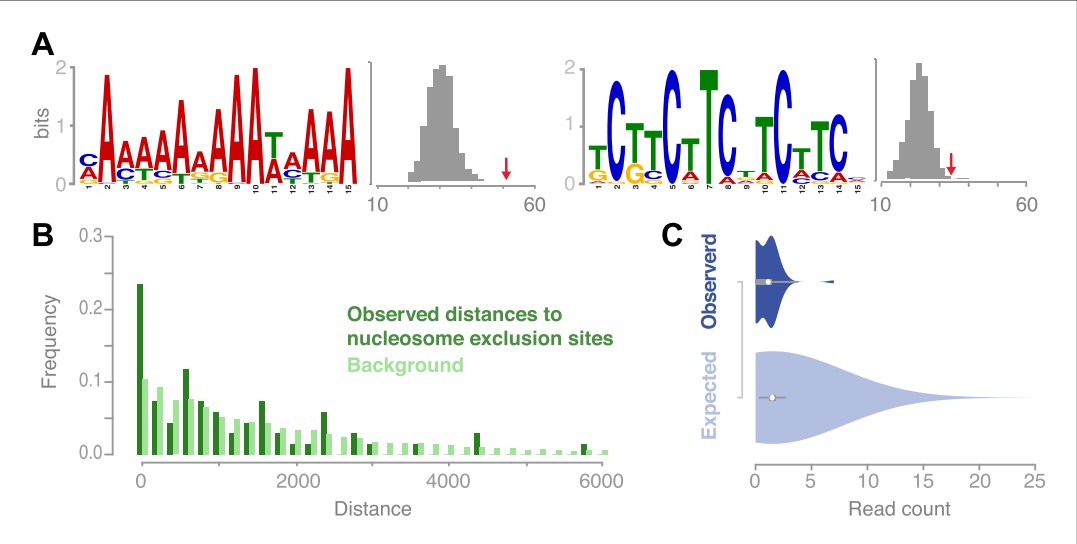

**Figure 5**. Recombination sites are significantly associated with two sequence motifs and nucleosome-free regions. (**A**) Two sequence motifs (poly-A and CTT/GAA) were found significantly enriched at recombination sites after searching for over-enriched motifs using MEME. We established background frequency distributions by randomly sampling regions of the same sizes from non-peri-centromeric regions, followed by a targeted search for the respective motif (shown as histograms). The observed number of motifs at recombination sites is shown by red arrows (poly-A: p value $3.8 \times 10^{-6}$; CTT: p value 0.002 [permutation test]). (**B**) The distances between recombination sites and DNA sequences that cannot be bound by nucleosomes are significantly enriched for short distances (p value 2.0e-16 [generalized linear model fitting]). Nucleosome exclusion sites are defined as $(A)_{10}$ and $((GC)_3NN)_3$ (Wang et al. (1996)) (**Suter et al., 2000**). (**C**) Comparison of recombination sites and nucleosome occupancy. The nucleosome occupancy was estimated through DNA sequencing performed after digesting with MNase of somatic tissue (**Chodavarapu et al., 2010**). The nucleosome-bound genomic regions are preferentially sequenced and establish a quantitative readout of nucleosome occupancy. The read count distributions at recombination sites (top) and at 5,000 random background regions (bottom) are significantly different (p value 1.9e-4 [*t* test]).

We examined known nucleosome exclusion sequences, that comprise poly-A (n≥10) and specific CG rich motifs ($[C/G]_3$-$N_2$-$[C/G]_3$-$N_2$-$[C/G]_3$) (**Wang and Griffith, 1996**; **Suter et al., 2000**) for their overrepresentation near recombination sites. The shortest distance from the recombination midpoints to the nearest nucleosome-exclusion motif was compared with a distance distribution based on randomly sampled sites (**Figure 5B**). Recombination events occurred much closer to nucleosome-exclusion sites than expected for random placement, which suggests that the *A. thaliana* recombination machinery targets constitutively open chromatin. To corroborate our hypothesis of recombination targeting open chromatin and being negatively correlated with nucleosome occupancy, we tested whether our recombination sites are known to be free of nucleosomes in somatic tissues. For this we compared the amount of DNA reads generated after digestion with MNase (**Chodavarapu et al., 2010**) for regions of conversion tracts to random non-peri-centromeric regions. Significant underrepresentation of DNA reads in conversion tracts corroborate that recombination is less likely in regions that are bound by nucleosomes (**Figure 5C**).

## Discussion

In order to describe the full extent of allelic exchange between *A. thaliana* homologous chromosomes, we have resequenced the complete genomes of 13 complete meiotic tetrads and 10 homozygous doubled haploid offspring of heterozygous $F_1$ hybrids. Genome rearrangements were shown to severely confound identification of genuine NCO–GCs, but stringent filters led to highly similar GC rate estimates for both offspring types. In particular, we showed how a complex 80 kb rearrangement led to the erroneous identification of recurrent double recombination events. Our results underscore the need for extreme caution and rigor when studying (biased) GCs using short reads.

## The rate of gene conversion in *A. thaliana*

*Sun et al. (2012)* estimated a GC rate of $3.5 \times 10^{-4}$ per marker per meiosis based on fluorescent reporter genes in pollen tetrads (*Francis et al., 2007*). This would translate to $8.8 \times 10^{-5}$ per site per meiosis. Since for most of their data these authors could not distinguish between NCO–GCs and CO–GCs, this frequency should be compared to the combined frequency of NCO–GC and CO–GCs in our study, which is $1.1 \times 10^{-5}$ per site per meiosis. Potential causes for differences may lie in local GC formation differences (*Sun et al., 2012*), experimental variation (e.g., plant growth conditions) or the higher probability of meiotic DSBs to occur in transgenes.

In *A. thaliana*, the ~1–3 NCO–GCs and ~10 COs per meiosis we observed do not amount to the 120 to 250 DSBs that were observed during meiotic prophase (*Vignard et al., 2007*; *Chelysheva et al., 2007*; *Sanchez-Moran et al., 2007*). Our simulation-based estimates of conversion tract lengths suggested short NCO tract lengths as a possible factor for this discrepancy. Small NCO tracts most often will not lead to conversions, as they do not overlap with polymorphisms. Minimal observed COCT lengths range from 1 to 1,229 bp (*Supplementary file 1F*), whereas the lengths of NCOCTs range from 1 to 282 bp (*Supplementary file 1H*). The recovery of two fairly long NCO events of 275 and 282 bp could be attributed to variation in NCO size, and is likely a detection bias, as NCOCT detection is strongly biased towards recovery of longer tracts (*Curtis et al., 1989*). A previous estimate of 558 bp for COCT length and less than 150 bp for NCOCT length (*Lu et al., 2012*) as based on six and four observations respectively, are close to our estimations. As suggested by *Lu et al. (2012)*, we cannot exclude other factors, like the preferential repair of heteroduplexes to parental genotypes (*Borts et al., 2000*) or inter-sister rather than inter-homologue repair to explain the low GC rate in *A. thaliana* (*Goldfarb and Lichten, 2010*). However, our data suggesting short NCO–GCs tracts do not require such additional assumptions.

Sequence identity at CO sites was as low as 82%, due to indels of up to 18 bp, which might not even comprise the lower limit of sequence divergence at CO sites. This may explain why previous work did not detect a relationship between CO frequencies and SNP density, albeit at a much coarser level (*Salome et al., 2012*).

Recombination events in *A. thaliana* are closely associated with nucleosome-free regions. The enrichment for poly-A motifs may explain this observation. Sequence annotation showed that promoter regions are enriched for recombination sites. These observations add to the emergent pattern that the recombination machinery targets accessible DNA at gene promoters in yeast and *Drosophila melanogaster* (*Pan et al., 2011*; *Comeron et al., 2012*). Even in mouse, where recombination hotspots are commonly guided by the DNA-binding methyltransferase PRDM9, gene promoters are targeted in the absence of PRDM9 (*Brick et al., 2012*). A GAA/CTT microsatellite motif was present at half of the investigated recombination sites in *A. thaliana*, and are (nearly) identical to two sequences found associated to ancestral hotspots (*Choi et al., 2013*). This motif is similar to the TRANSLOCON 1 (TL1) binding motif, to which the heat-shock factor-like transcription factor HSFB1 binds (*Pajerowska-Mukhtar et al., 2012*; *Wang et al., 2005*). There are however no indications that this transcription factor is involved in *A. thaliana* meiosis.

## The impact of recombination on population-wide allele frequencies

To understand the coherent flow of genes and alleles within natural populations of *A. thaliana* (*Bomblies et al., 2010*; *Cao et al., 2011*; *Long et al., 2013*) it is essential to understand not only the selective forces that affect allele frequencies, but also the mechanisms that introduce and distribute genetic variation. Together with the random inheritance of homologous chromosomes, COs are the major factors generating new allele combinations by redistributing allelic variation through exchanging complete chromosome arms. The prevalent co-conversion of polymorphisms at CO sites prohibits the formation of complex crossover patterns of alternating polymorphisms, which implies that it is the COs themselves that diffuse haplotype borders, rather than their associated gene conversions.

NCO–GCs exchange variation between existing haplotypes and as such can effectively generate new alleles of genes. We recovered four NCOs that indeed lead to the generation of new allelic variants of genes through NCO. However, GCs can only generate new allele combinations in heterozygous plants. In the case of *A. thaliana*, which has a selfing rate of at least 85% (and much higher in most populations [*Bomblies et al., 2010*]), this impact will be relatively small in comparison to other organisms.

As shown in our analyses, meiotic recombination is an effective mechanism to generate new copy number variants, as was previously suggested by *Lu et al. (2012)*. We described how CO events altered the copy number of the transposed sequence within the offspring genomes. The identification of an 80 kb transposition that is present on the same chromosome, but distant enough to allow for intermittent CO events, generated haploid genotypes that either lost or duplicated the transposed sequence. While we can currently only speculate on the resulting fitness (dis-)advantages for their offspring, the genomic effect of combining crossover recombination with genomic structural variation (*Schneeberger et al., 2011*; *Gan et al., 2011*; *Schmitz et al., 2013*; *Long et al., 2013*; *Cao et al., 2011*) provides significant potential for further selection and shaping of the *A. thaliana* pan-genome.

## Materials and methods

### Generation of meiotic tetrads and doubled haploids lines

For the generation of meiotic tetrads, a Col–L*er* hybrid in a *quartet*1 background was made by crossing *qrt1 −/−* Col (N660403) to *qrt1 −/−* L*er* (N8050), which were obtained from the Nottingham *Arabidopsis* Stock Center. Single meiotic pollen quartets from this F$_1$ plant were picked up with a hair under a microscope and transferred onto a virgin flower of a male sterile Cape Verde Islands (Cvi) female receptor. The male sterility mutant was selected from an EMS treated Cvi population and backcrossed twice to Cvi. These plants were grown under standard long day conditions in a growth chamber. Over 700 unique pollinations were made. All resulting siliques were individually harvested, and when four seeds were recovered from one silique, the resulting plants were grown under short day conditions to maximize rosette size before harvesting. Plants were genotyped using a previously described SNP marker set (*Wijnker et al., 2012*) to verify that all markers segregate in the expected tetrad 2:2 ratio. Doubled haploid *Arabidopsis* lines were selected from crosses made for an existing Columbia (Col)—L*er* DH population. Five of these DH1, DH2, DH3, DH5 and DH10 were featured in a previous publication as plants 53, 34, 58, 49 and 72 respectively (*Wijnker et al., 2012*). The other five DH lines were selected from among (doubled) haploid lines that selected from the same crosses after initial publication of this material.

### Library preparation and sequencing

DNA of all five parental lines, DHs and tetrad offspring was extracted from adult rosettes using the CTAB method, with a nuclei extraction step to remove mitochondrial and chloroplast DNA (*Cao et al., 2011*). One-half to one gram of *A. thaliana* leaves were ground to a fine powder in liquid nitrogen using mortar and pestle and transferred to a 15-ml polyethylene centrifuge tube containing 10 ml of ice-cold nuclei extraction buffer, consisting of 10 mM TRIS-HCl pH 9.5, 10 mM EDTA pH 8.0, 100 mM KCl, 500 mM sucrose, 4 mM spermidine, 1 mM spermine and 0.1% beta-mercaptoethanol. The suspended tissue was mixed thoroughly with a wide-bore pipette and filtered through two layers of Miracloth (CalBiochem, San Diego, CA) into an ice-cold 50-ml polyethylene centrifuge tube by a brief spin at less than 100 g for 5 s. Two ml lysis buffer, consisting of 100 mM Tris ph7.5, 0.7M NaCl, 10 mM EDTA, 1% BME (2-mercaptoethanol) and 1% CTAB in H$_2$O, were added to the filtered suspension and mixed gently for 2 min on ice. The nuclei were pelleted by centrifugation at 2000 g for 10 min at 4°C. 500 ul CTAB extraction buffer was added to the nuclei pellet, mixed well by inverting the tube and incubated for 30 min at 60°C. The sample was then cooled for 5 min at RT before adding 350 μl Chloroform/isoamyl alcohol (24:1), inverting and mixing gently for about 5 min, and spinning in a microcentrifuge at 6000 rpm for 10 min. The upper layer (450 μl) was transferred to a new 2-ml tube containing 450 μl isopropanol and mixed by inverting several times before pelleting the DNA by centrifugation at 13000 rpm for 3 min. After washing the DNA pellet in 75% EtOH, the DNA was resuspended in sterile DNase free water (containing RNaseA 10 μg/ml). The sample was incubated at 65°C for 20 min to destroy any DNases, and stored at 4°C until use. The DNA concentration and quality was determined with a Nanodrop 1000 (Peqlab, Erlangen, Germany), a Qubit 2.0 Fluorometer (Life Technologies, Carlsbad, CA, USA) and on a 1% agarose gel. DNA samples were concentrated to more than 50 ng/μl with a speed-vac when necessary.

The DNA samples were sequenced by the Max Planck Genome Center Cologne, Cologne, Germany, and by the Max Planck Institute for Developmental Biology, Tübingen, Germany. At both sequencing facilities, quality checks were performed with a Bioanalyzer (Agilent 2100, Agilent, Böblingen, Germany). Libraries for the doubled haploids, parents and the five deeply sequenced tetrads were

generated using the Illumina Genomic DNA TruSeq sample kit (Illumina, San Diego, CA) according to the manufacturer's instructions. Libraries for the eight shallowly sequenced tetrads were prepared by fragmenting the DNA using dsDNA Shearase (Zymo Research, Irvine, CA, USA) to obtain 100- to 1000-bp fragments, A-tailing using Klenow exonuclease (New England Biolabs, Ipswitch, MA, USA), ligating to indexed adapters using the QuickLigation kit (New England Biolabs, Ipswitch, MA, USA), size selection to 300 to 500 bp using AMPure XP SPRI beads (Beckman-Coulter, Pasadena, CA), and PCR enrichment using the Phusion DNA polymerase (New England Biolabs, Ipswitch, MA, USA). The complete methodological details for the latter library preparation protocol will appear in a separate manuscript that is currently in preparation for publication. The samples were sequenced on Illumina Genome Analyzer GAIIx and Illumina HiSeq2000 in 100 bp paired-end runs. Sequencing yield per sample is listed in *Supplementary file 1A*.

### Resequencing analysis

We applied SHORE to the whole-genome sequencing data of the five parental accessions (parents of tetrads: *qrt* Col, *qrt* L*er* and EMS male sterile Cvi, and the Col and L*er* parents of the DH lines), the 52 individual tetrad genomes and the 10 DH offspring independently (*Ossowski et al., 2008*). For each sample, short reads were quality filtered and trimmed using the default values. High quality reads were then aligned against the Arabidopsis reference sequence using GenomeMapper by allowing up to 10% mismatches and gaps (*Schneeberger et al., 2009*; *Lamesch et al., 2012*). After using read pair information to remove repetitive alignments, we performed consensus calling again using the default parameters.

### Initial marker definition and CO identification for tetrad analysis

Initial markers included all loci with a quality score greater than 24 in the analysis of Col-0 and Cvi, and a quality score of 40 (which is the maximum SHORE assigns) within the analysis of L*er*. In addition we removed all markers that resided in putative duplications, transposable elements, in regions that showed enriched coverage or that were closer than 150 bp to a position that showed evidence of two different alleles in the resequencing of one of the parental genomes (*Supplementary file 2A*).

For the reconstruction of the individual tetrad genomes, we assigned either a Col or L*er* allele to each marker if there was a resequencing quality score greater than 15. This initial genotyping was used to identify COs by merging consecutive markers of the same genotype into blocks of markers. Blocks with at least 25 consecutive markers were used as seeds. COs positions were identified by extending the seeds until the nearest block of the other genotype with more markers than the preceding block.

### Identification and rate estimation of NCO–GCs in tetrad genomes

NCO–GC identification was attempted in the deeply sequenced tetrad genomes only. An NCO–GC was reported if the genotype at a marker was different from the background that was assigned in the initial genotyping. In order to distinguish between homozygous and heterozygous genotypes reliably, we required a minimum coverage threshold and a coverage-dependent allele frequency cutoff. We established a coverage-dependent frequency cutoff that allows identifying homozygous and heterozygous genotypes at identical error rates. To this end, we fitted beta distributions to the observed distribution of allele frequencies as measured by the short read alignments at type-1 and type-2 markers respectively (an example for a minimal coverage of 50 is shown in *Figure 2—figure supplement 2*). The unique point at which both distributions share the same quartile has been defined as the allele frequency threshold. This was repeated at multiple minimal coverage thresholds. Assigning homozygous and heterozygous genotypes at the same error rate allows for the calculation of the frequency of NCO–GCs by dividing markers with assigned NCO–GC by all markers with an assigned genotype. For the identification of a minimal coverage threshold, NCO–GC frequencies were calculated at multiple minimal coverage values. Calculations with lower coverage thresholds revealed higher NCO–GC rates compared to calculations with more stringent coverage thresholds (*Figure 2—figure supplement 3*).

NCO–GC identification was initially performed on the set of markers used for CO identification. Though this marker set was defined in order to include high quality markers only, further filtering was necessary. Several filtering steps were applied, each reducing the number of markers. First, we extended the regions which encompass putative duplications and which were not allowed to feature markers. Each transposable element and putative duplicated region (as defined by the resequencing analysis performed with SHORE) was extended by 2 kb. Regions around heterozygous positions were

defined as 1 kb up- and down-stream. The second filtering excluded all those markers in which we found evidence of read pair alignments from three different parents, which is a reliable indicator of wrong alignments. As a third filtering step, we removed markers where local sequence divergence hampered the alignment of Ler short reads to the reference sequence.

Finally we removed all markers that were shown to segregate in a 2:2 manner within the Sanger sequencing data. As the absolute number of false positive NCO–GCs was relatively low compared to the absolute number of markers, we conclude that this is also true for the non-NCO–GCs. Errors result from technical artifacts and thus affect NCO–GCs and non-NCO–GCs equally, and thus the absolute number of false negatives is negligible.

## Identification and validation of the 80 kb of transposed sequences

Initial identification of the 80 kb transposition was based on the recombinant tetrad genomes showing copy number variation at the transposition sites, which led to false GC calls. Local *de novo* assembly of reads aligned to this region and prior access to a *de novo* assembly of the Ler genome, which will appear in a separate manuscript that is currently in preparation for publication, revealed the insertion sites and allowed for primer design. *Figure 2—figure supplement 5* illustrates the validation of transposed and inverted regions through PCRs. These PCRs were done on the Col and Ler parental lines. *Supplementary file 2F* lists used primer sequences.

## Refinement of CO and NCO conversion tracts

In order to get a complete picture of the polymorphisms in the COCT and NCOCT in both the tetrad and the DH samples, we manually parsed all short read alignments around CO and NCO–GCs combined with local short read assemblies of the short reads aligning to the respective region in addition to the read mate pairs that were not aligned using Velvet (*Zerbino and Birney, 2008*).

## Validation of NCO–GCs in tetrads

Putative NCO–GC events were validated by Sanger sequencing of PCR products of all four tetrad offspring. All PCRs were done on extracted genomic DNA, except for NCO–GC events in tetrad 58, where DNA of the sequencing library was used. *Supplementary file 2D* lists primer sequences for all NCOs.

## Recombination interference

Distances between neighboring COs and NCOs events were calculated using the midpoint of each CO and NCO as its unique location. To assess whether interference acts on different types of recombination events we implemented a permutation test for adjacent recombination events as suggested by *Mancera et al. (2008)*. For each permutation test we generated 10,000 iterations. Instead of randomizing recombination locations we randomized the labels of the tetrads of each CO, and thereby preserved the inhomogeneity of CO placement across the genome. Chromatid interference, the non-random association of the chromatids involved in two adjacent CO, would violate the validity of this background distribution of our permutation tests. However, we could not identify any evidence for chromatid interference in our data.

## Validation of NCOs in DH lines

Putative NCO–GC events were validated by Sanger sequencing of PCR products of both parental lines Col and Ler and each respective DH line. *Supplementary file 2E* lists used primer sequences for each NCO.

## Marker definition and genotyping of DH samples

High quality markers were defined as positions with a quality score of 25 or greater in the resequencing of the Col-0 sample, a homozygous SNP in the resequencing analysis of Ler with a quality value of 40 and a short read coverage between 50 and 150 read alignments for the resequencing of Ler. We genotyped each of the 10 DH samples at each of the markers, when we identified a homozygous consensus call with a quality score of 15 or more. Initial genotyping of the 438,919 markers and CO location identification was performed as outlined for the tetrad genomes.

NCO–GCs are identified at markers with genotypes that differ from the expected parental genotype. We used the coverage-dependent frequency cutoffs identified in the analysis of the tetrad samples in order to identify to what level of ambiguous reads can be accepted at homozygous positions. This allowed the identification (and removal) of markers that are most likely featuring alignments from

different loci in the genome. Calculating the probability of NCO–GCs per marker at a series of minimum coverage thresholds revealed that beyond a minimal coverage of 10, the frequency of NCO–GC remained stable. This threshold was used in the subsequent identification of GCs.

## Identification of spontaneous mutations in the DH lines

For the sequencing of the doubled haploid samples, we pooled the DNA of at least four sibling double haploid plants for each sample. Spontaneous mutations, which occurred in the pollen or egg cell of the haploid progenitor plant, as well as recent somatic mutations of individual doubled haploid plants will not be fixed in these pools. By comparing the resequencing data of all 10 samples, we identified mutations, which are specific for one of the DH samples. These encompass mutations that happened in the haploid life cycle before meiotic cell divisions. Only homozygous positions with a SHORE score of greater than or equal to 32 were probed for spontaneous mutations. Like the identification of NCO–GCs, the identification of spontaneous mutations required a minimal coverage of 10, an allele frequency higher than the cutoff to distinguish between homozygous and heterozygous positions (as defined for the tetrad analysis), and equivalent evidence for the non-mutant allele in the other samples.

## Meiotic DSB-associated motif identification

We searched for consensus motifs at 67 conversion tracts identified in tetrads and DH lines. Performing motif searches on conversion tracts ensures that the sequences were subjected to DSB repair. However, as conversion tracts are not necessarily centered on the location of the respective DSB, we included flanking sequences of 500 bp to increase the probability to encompass the complete region targeted by DSB repair. Candidate motifs were identified with MEME (*Bailey et al., 2006*), which was run with the 'zoops' model, while correcting for the genomic background. Motifs with a minimum of five and maximum of 15 bp were identified. Only two motifs featured an e value of less than 1e-05. Due to the repetitive nature of the recovered motifs, we performed an additional permutation to test for random occurrence in the genome. Rescreening was performed within the observed conversion tracts as well as in 1,000 random genomic region of the same length using a positional weight matrix. The positional weight matrix was mapped against each sequence using MOODS and matches with p value <0.001 were considered (*Korhonen et al., 2009*).

## Data deposition

Short read data have been deposited in the EBI short read archive under accession number ERP003793.

# Acknowledgements

Melany Bartsch and Wim Soppe are thanked for providing the $M_2$ population derived from EMS-treated Cvi Jose van de Belt and Marijke Hartog are acknowledged for their help with PCRs.

# Additional information

### Competing interests

DW: Deputy Editor, *eLife*. The other authors declare that no competing interests exist.

### Funding

| Funder | Author |
| --- | --- |
| Gottfried Wilhelm Leibniz Award from the Deutsche Forschungsgemeinschaft | Detlef Weigel |
| Alexander von Humboldt Foundation | Beth A Rowan |
| Max Planck Society | Geo Velikkakam James, Jonas R Klasen, Vimal Rawat, Korbinian Schneeberger |
| EMBO long-term fellowship | Erik Wijnker |

The funders had no role in study design, data collection and interpretation, or the decision to submit the work for publication.

## Author contributions
EW, JJBK, Conception and design, Acquisition of data, Analysis and interpretation of data, Drafting or revising the article; GVJ, JD, JRK, VR, Analysis and interpretation of data, Drafting or revising the article; FB, DFJ, CBS, BH, Acquisition of data, Drafting or revising the article; BAR, Acquisition of data, Analysis and interpretation of data, Drafting or revising the article; LZ, SO, Analysis and interpretation of data, Drafting or revising the article, Contributed unpublished essential data or reagents; HJ, DW, MK, Conception and design, Drafting or revising the article; KS, Conception and design, Analysis and interpretation of data, Drafting or revising the article

## Additional files

### Supplementary files
• Supplementary file 1. (**A**) Read number. (**B**) Number of crossovers per tetrad and DH line and chromosome. (**C**) CO location. (**D**) NCO location and extent within the tetrad samples. (**E**) CO location in DH lines. (**F**) Exact loci of CO and extent of associated GC in tetrad samples. (**G**) Converted markers within COCTs in tetrad samples. (**H**) Converted markers within NCOCTs in DH lines.

• Supplementary file 2. (**A**) Initial marker list used for an initial genotyping of the tetrad samples. (**B**) Filtered marker list used to genotype tetrad samples. (**C**) High quality marker list used to genotype DH samples. (**D**) Primer pairs used for verification of NCOs in tetrad offspring. (**E**) Primer pairs used for verification of NCOs in DH lines. (**F**) Primers used for the verification of transposed DNA on chromosome 3.

• Supplementary file 3. Includes a visualization of the exact makeup of 71 COs identified in the tetrad samples, for which sufficient sequencing information was available. The colored areas indicate the number of short read alignments for each of the positions as indicated on the x-axis. Red and blue areas refer to regions that descended from Col-0 and Ler, respectively. Grey areas cannot be assigned to either of them. Vertical lines indicate sequence differences between the parental genotypes and have the respective genotype indicated next to them. Within the CO sites (between the outer borders of both grey areas) the polymorphism data are based on hand curated short read alignments and local assemblies. Outside the flanking markers the polymorphisms encompass the marker used for reconstructing the recombinant chromosomes only. Note each tetrad sample contains one recombinant chromosome and one that is derived from the Cvi parent. The Cvi alleles are not indicated in these plots.

### Major dataset
The following dataset was generated:

| Author(s) | Year | Dataset title | Dataset ID and/or URL | Database, license, and accessibility information |
|---|---|---|---|---|
| Wijnker E, Velikkakam James G, Ding J, Becker F, Klasen JR, Rawat V, et al. | 2013 | *Arabidopsis thaliana* recombinant tetrads and DH lines | ERP003793; http://www.ebi.ac.uk/ena/data/view/ERP003793 | Publicly available at the European Nucleotide Archive (http://www.ebi.ac.uk/ena/). |

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
