## [Decision Letter]

Thank you for sending your work entitled “The genomic landscape of meiotic crossovers and gene conversions in *Arabidopsis thaliana*” for consideration at *eLife*. Your article has been favorably evaluated by a Senior editor and 3 reviewers, one of whom is a member of our Board of Reviewing Editors.

The Reviewing editor and the other reviewers discussed their comments before we reached this decision, and the Reviewing editor has assembled the following comments to help you prepare a revised submission.

The distribution of non-crossover association gene conversion (NCO-GC) events in large genomes (such as those of plants and mammals) is of interest to those studying both recombination processes and also the evolutionary impacts of recombination. Several previous papers have studied rates and patterns of NCO-GC in Arabidopsis. This paper adds to the literature by sequencing both meiotic tetrads (13 total, 5 in great detail) and doubled haploids (DH; 10 total). The authors take great care to reduce the false positive rate for gene-conversion events, which results in a fairly limited number of events (7 validated NCO-GC events meeting all filters from the tetrads, 9 from the DH lines). They analyse the rate of GC (both CO and NCO), tract lengths, interference, relationship to genomic feature and possible motifs. Although the findings largely replicate those seen in other organisms, the demonstration in Arabidopsis is important and the experimental is carried out and analysed at a very high level.

In revision, we would like you to address the following points.

1) It would be interesting to see a blow-up of any CO events associated with NCOs with patterns of Col and Ler markers between the 4 sister genomes shown. In particular, to show the spatial relationships between the observed COs and gene conversion tracts.

2) When trying to fit models of tract length and GC rate, it is important to include not just the observed number of conversion events, but also the distribution of the number of SNPs co-converted. This will help resolve the relationship between rate and length.

3) During sequencing of the DH lines the authors observe a two-fold higher rate of mutation compared to published rates, which is attributed to their haploid life cycle. Do the authors mean that this occurs due to the absence of a sister template to repair from? Interestingly, they estimate a higher conversion frequency at COs – do the authors have enough information to know whether specific types of mutations (e.g., transitions/transversions) are occurring?

---

## [Author Response]

*1) It would be interesting to see a blow-up of any CO events associated with NCOs with patterns of Col and Ler markers between the 4 sister genomes shown. In particular, to show the spatial relationships between the observed COs and gene conversion tracts*.

We have added visualizations of the genomic regions of 71 CO sites, for which sufficient sequencing information was available for the reconstruction of putative conversions tract. The visualizations include the parental genomes and all four genomes of the respective tetrads (see Supplementary file 3).

*2) When trying to fit models of tract length and GC rate, it is important to include not just the observed number of conversion events, but also the distribution of the number of SNPs co-converted. This will help resolve the relationship between rate and length*.

We agree this indeed advances our simulations. We have now included a comparison of the average number of co-converted markers per conversion tract. In addition, we do not only show average values, but added standard deviations. This additional analysis corroborates that the tract lengths of COs and NCOs are drastically different.

*3) During sequencing of the DH lines the authors observe a two-fold higher rate of mutation compared to published rates, which is attributed to their haploid life cycle. Do the authors mean that this occurs due to the absence of a sister template to repair from*?

The higher mutation rate in Arabidopsis haploids may have diverse causes. The absence of a homologous repair template in haploid cells in G1 might lead to more (presumably error-prone) repair through non-homologous end-joining (NHEJ). However, it is also conceivable that the process of uni-parental genome elimination causes mutagenic stress (i.e., see [51], PNAS). We have now added a sentence that clarifies this. In the original version of the manuscript we included 1-bp deletions in the calculation of the mutation rates. Excluding those (as it was done in the estimation of the wild-type mutation rate ([40], Science)) decreased our estimation slightly, but it is still more than 1.5 times higher as compared to published rates. Like for the original calculations this observation does not reveal a statistically significant difference.

Independent of any mechanistic reason for a putative enrichment in spontaneous mutations we had to check whether the mutation frequency is elevated to such drastic degrees that it would interfere with the identification with NCO-GCs. This was not the case.

*Interestingly, they estimate a higher conversion frequency at COs – do the authors have enough information to know whether specific types of mutations (e.g., transitions/transversions) are occurring*?

Indeed, our analysis (of the tetrad samples) suggests higher conversion frequencies at COs as compared to NCOs, however this difference is not significant. The overall limited number of spontaneous point mutations makes it hard to characterize the spectrum of mutations. However among these eight mutations were 6 transitions and 2 transversions. This fits to the enrichment of transitions among spontaneous mutations, which was reported earlier ([40], Science). We mention this in the updated version of the manuscript.